# Toward the Specificity of Bare Nanomaterial Surfaces for Protein Corona Formation

**DOI:** 10.3390/ijms22147625

**Published:** 2021-07-16

**Authors:** Fabio Vianello, Alessandro Cecconello, Massimiliano Magro

**Affiliations:** Department of Comparative Biomedicine and Food Science, University of Padua, Viale dell’Università 16, 35020 Legnaro, Italy; fabio.vianello@unipd.it (F.V.); alessandro.cecconello@gmail.com (A.C.)

**Keywords:** nanoparticles, biomolecules, protein corona, uncoated nanomaterials, surface recognition

## Abstract

Aiming at creating smart nanomaterials for biomedical applications, nanotechnology aspires to develop a new generation of nanomaterials with the ability to recognize different biological components in a complex environment. It is common opinion that nanomaterials must be coated with organic or inorganic layers as a mandatory prerequisite for applications in biological systems. Thus, it is the nanomaterial surface coating that predominantly controls the nanomaterial fate in the biological environment. In the last decades, interdisciplinary studies involving not only life sciences, but all branches of scientific research, provided hints for obtaining uncoated inorganic materials able to interact with biological systems with high complexity and selectivity. Herein, the fragmentary literature on the interactions between bare abiotic materials and biological components is reviewed. Moreover, the most relevant examples of selective binding and the conceptualization of the general principles behind recognition mechanisms were provided. Nanoparticle features, such as crystalline facets, density and distribution of surface chemical groups, and surface roughness and topography were encompassed for deepening the comprehension of the general concept of recognition patterns.

## 1. Introduction

Although the constant improvements in nanoscience and nanotechnology are raising the bar of nanomaterial applications in biomedicine, the efforts of manufacturers are very often frustrated by the adverse effects of biological environments on nanomaterials. The main reason is that, upon exposure to a biological environment, the nanomaterial surface is subjected to competitive binding of different macromolecules, resulting in a biological shell, called protein corona, characterized by a composition that can be extremely variable depending on biological milieus and time. As a consequence, upon their introduction in vivo, engineered nanoparticles enter into an inevitable and hardly predictable progressive transformation that, in most cases, leads to their elimination.

The in-depth comprehension of the interactions and mutual influences between proteins and nanoparticles represents one of the main goals for the successful transfer of nanotechnologies into medicine [1,2] and, more specifically, for the control over protein corona formation in vivo, which can be pivotal for determining opposite scenarios: the elimination or the effective internalization of nanocarriers [3].

In recent years, attempts to modulate the nature of the protein corona were carried out by designing tailored nanoparticle ligands and their distribution on the nanomaterial surface [4,5,6,7,8]. Specific ligands can ideally control the nature of the protein shell by the incorporation of suitable functionalities for the selection and orientation of proteins during the formation of soft or hard corona shells [9]. As an example, hydrophobic coatings on gold nanoparticles (AuNPs) induce protein denaturation and irreversible protein binding (hard corona), making proteins readily susceptible to proteolysis by endogenous enzymes [10]. Conversely, AuNPs coated with tetra(ethylene glycol) (TEG), which prevents hydrophobic interactions, formed soft coronas with retained protein structure while slowing proteolysis [9,11,12]. Furthermore, the zwitterionic sulfobetaine surfactant has been shown to provide “stealth” properties to AuNPs, leading to corona-free nanoparticles in serum [5]. In contrast, positively charged quaternary ammonium groups were shown to interact with proteins, readily forming protein coronas on NPs [6]. On this bases, surface modifications of nanomaterials would seem to be the only viable way for exerting a control over the protein corona formation. As this review will attempt to highlight, this is not entirely true, and a deeper comprehension should be dedicated to protein corona formation on bare nanomaterials.

In our opinion, a novel set of opportunities could be at hand by filling the gap between nanomedicine, still suffering from a lack of comprehension regarding the interplay between proteins and nanoparticles, and the understanding of interactions between biotic-abiotic materials learned by other scientific areas. In fact, an impressive body of studies, spanning from bio-mineralization to prebiotic chemistry, suggests that the interaction of proteins with bare solid surfaces, such as inorganic nanomaterials, can be highly sophisticated and characterized by high selectivity. In this view, the idea of surface coating as an unavoidable prerequisite, mandatorily and indiscriminately applicable to any kind of nanomaterial, would preclude the actual application of bare nanomaterials in biomedicine, thus depriving this scientific branch of opportunities given by this class of nanomaterials.

It should be considered that the surface of minerals played a crucial role in the chemical evolution of the prebiotic era and in the origin of life. Indeed, reactions at the mineral-–water interface were proposed for the initial synthesis of biomolecules [13]. Thus, to living (biotic) systems inorganic (abiotic) materials are not alien.

The present review, aimed at summarizing the principles governing the interactions between proteins and uncoated nanoparticles, pays particular attention to the abiotic–biotic interface and tries to integrate insights of scientific research involving prebiotic chemistry, biomineralization, chiral recognition, protein purification, and material science. Thus, hints from different contexts were reviewed and compared in order to provide the reader with a general view of the principles controlling protein-(bare) nanoparticle recognition. In this view, a variety of nanomaterial features were highlighted, such as crystalline structure and facets [14], density and distribution of surface chemical groups [15], and surface roughness and topography [16]. Furthermore, synthetic nanomaterials displaying selectivity toward biomolecules were reviewed.

Beyond applications in nanomedicine, an overall vision on the interactions between proteins and bare nanomaterials is here presented, possibly offering a perspective for predicting the formation, the stability, and the biological properties of protein coronas. Considering the magnitude of the topic, the present review does not claim to be comprehensive. On the contrary, it is aimed at stimulating the reader by summarizing selected points from apparently distant contexts.

## 2. The Role of Protein Selectivity in Biomedical Applications of Nanomaterials

Upon exposing a nanomaterial to a biological fluid, a protein corona shell is generated. Depending on the environment, an early-stage corona (soft corona) is gradually replaced by a shell comprising proteins that display a higher affinity for the nanoparticle surface (hard corona). This competitive displacement of earlier and less specific proteins is named “Vroman effect,” resulting in undesired layer instability which can be hardly controlled [17]. Thus, the nanostructure surface is destined to evolve into a complex biological interface which is actually what the organism “sees” once the synthetic nano-object is introduced into the body [3]. It is well known that protein corona composition and structure are influenced by the size, shape, and chemical composition of the nanomaterial surface. In addition, the specific environment the nanoparticles are exposed to and the duration of the exposure play fundamental roles in the process [18]. Indeed, exposure time is crucial due to the aforementioned dynamic nature of the interaction [19], making the prediction of the corona composition and its biological properties even harder [4,20,21]. A further critical factor is protein modification induced by interactions with the nanomaterial surface. These structural alterations will eventually result in decreased protein flexibility and loss of biological activity. In the worst situation, the adsorption phenomenon leads to protein denaturation. Numerous studies that focused on the description of binding modes and the orientation of adsorbed proteins on nanomaterials, stressed the importance of the control of the biotic–abiotic interactions. As an example, it is crucial to ensure that the catalytic site of an enzyme remains accessible after adsorption on the nanomaterial, since steric hindrance and conformational changes are likely to affect biological activity [22,23,24,25]. Again, proteins are flexible objects, and their internal dynamics play a central role in their biological function [26,27]. This last issue further exacerbates the complexity associated with designing nanomaterials, which ideally should be built for the binding of specific proteins. On this basis, for several decades protein corona was considered responsible for the triggering of the immune system and the final clearance of the nanomaterial. Actually, for most cases, protein corona formation is the very first step in the process of nanoparticle elimination [28]. Generally, coronas formed from the unspecific protein adsorption act as labels promoting nanoparticle recognition as “non-self” and, as a consequence, nanoparticle elimination by the immune system [29,30]. As a consequence, for a long time research has been focused on the development of antifouling coatings aimed at the complete suppression of biomolecule binding to nanomaterials. Noteworthy, antifouling coatings were believed to be the only coatings providing nanomaterials with the ability to avoid the immune system clearance. This phenomenon has been named the “stealth” effect and bio-inert coatings have been defined as “stealth materials.” Among them, the gold standard is represented by polyethylene glycol (PEG).

Paradoxically, “stealth” materials present immunological drawbacks. For example, it is well-known that the prolonged use of PEGylated materials results in the production of anti-PEG IgM antibodies [31,32] in a phenomenon known as “accelerated blood clearance” (ABC) [33]. Thus, with time, PEGylated nanocarriers are recognized by IgMs and accumulated in the liver [34]. Moreover, these drug delivery stealth vectors, although providing advantageous features such as long-term circulation in the blood, are in turn detrimental for cell targeting, internalization, and accumulation [35]. Complete protein corona suppression, besides being an unreachable goal, was recently questioned by the demonstration that the “stealth effect” requires the binding of specific plasma proteins. Thus, a nanomaterial, in order to avoid the immune system reaction, does not need to be protein repellent but protein selective. Noteworthy, such a revolutionary principle was demonstrated by studying PEG-modified nanoparticles [36]. Indeed, the concept of controlling the phenomenon of protein corona formation by designing novel and specific nanoparticle surfaces is progressively replacing the illusion of zeroing macromolecular adsorption [37]. Thus, elucidating the molecular interactions at the nano-bio interface will be crucial for predicting protein corona formation and composition of designed nano-vectors characterized by high biocompatibility and specific targeting functions [38,39].

An ideal protein corona should act as camouflage, promoting nanoparticle recognition as “self” by the organism and, at the same time, acting as a chaperone, promoting the crossing of biological barriers (Figure 1). In this view, the merging of research from different fields of science is becoming increasingly urgent in order to bridge in vitro technologies with in vivo biomedicine. Moreover, synthetic biology will benefit from the comprehension of interactions occurring between biomolecules (biotic) and materials (abiotic) at the molecular level [40]. In this context, the concept of “epitope map,” proposed by Dawson and colleagues about twenty years ago [41], aimed at mapping the actual surface of the protein corona, ideally encompassing both protein composition and conformation. The epitope map is seen as the “biological fingerprint” of the particle–protein complex and defines its biological activity. Thus, the outer layer of the biomolecular envelop controls the interaction of the nanoparticles with cells and the nanoparticles trafficking to different cell compartments. Hence, this biological “mask” determines the fate of synthetic nanoparticles in biological systems.

Among synthetic uncoated nanomaterials, carbon-based nanomaterials (CNMs) can count on a notable number of in vitro cell studies dealing with fundamental aspects, such as cell uptake, cytotoxicity, and drug delivery and, most importantly, providing valuable hints on the relationships linking nanomaterial surface, biomolecular interaction, and cell response. Thus, herein CNMs will be discussed first because, besides being at an advanced stage from the standpoint of biomedical applications, they represent an optimal example of how nanomaterial syntheses and post-synthetic modifications can modulate the selectivity of a bare surface toward proteins and stimulate, in this way, different cell responses.

## 3. Carbon-Based Nanomaterials

Surface features and defects are of great importance for the recognition of inorganic nanomaterials (*vide infra*) by peptides and proteins, and this paradigm was also confirmed for carbon-based nanomaterials. Sengupta and colleagues showed that in CNMs superficial defect-induced hydrophilicity strongly influenced the formation of protein coronas composed of albumin and fibrinogen [42].

Due to their unique physicochemical properties, among CNMs, carbon nanotubes (CNTs) aroused great interest for their potential application in cancer treatments as nanocarriers for drugs [43]. Besides their high surface-to-volume ratio, CNTs display delocalized π-electron clouds which are believed to promote protein corona formation. This peculiar feature has a potential application in antigen delivery and could be used for the development of strong and long-lasting antigen-specific immune responses. Bai and colleagues reported on carboxylated multiwalled CNTs (MWCNTs) as advanced carriers to deliver ovalbumin [44]. Combinatorial phage-display methods were successfully applied for selecting specific amino acid sequences interacting with a plethora of nanomaterials [45]. For example, histidine- and tryptophan-rich protein motifs showing a selective affinity for carbon nanotubes were identified using phage-display technology. Intriguingly, these binding sequences were flexible and folded into a structure matching the geometry of CNTs [46]. Primary, secondary, and tertiary structures of peptides or proteins are believed to play a crucial role in their binding to CNTs [47], following a model based on shape complementarity. For carboxylated MWCNTs, it was found that the higher the defect density on the nanomaterial surface, the greater the ovalbumin adsorption and consequent macrophage activation, indicating a new fascinating option for the development of next-generation vaccines. Again, it appears that the interaction of CNTs with biomolecules can be modulated by the introduction of structural defects. As reported in the ex vivo study by Raghavendra and colleagues, this can be obtained by mechanical stress, where protein corona composition on single-walled CNTs (SWCNTs) was evaluated as a function of the physicochemical alterations of the nanomaterial obtained by ball-milling. Furthermore, the phenomenon of protein corona formation was responsive to proteome variations and to the biological milieu [48].

Machova and colleagues presented a systematic in vitro study on protein corona development using ultra-small (2 nm) hydrogenated or oxidized nano-diamonds, elucidating their interaction with human cells. The hydrogen- or oxygen-rich surfaces emerged as important discriminating factors orienting the surface chemistry toward protein selection. Different and specific protein coronas influenced the interactions of nano-diamonds with cells, resulting in different levels of cytotoxicity [49].

Graphene-based materials (GBMs) represent other objects that are the focus of intense studies, due to their unique tunability, physico-chemical and surface properties, and great potential for biomedical applications. These are far from being fully exploited, and there is still much to be understood in the context of graphene/biomolecular interactions/cell responses. For example, graphene has been demonstrated to stimulate different cell responses and to influence the interactions with proteins in multiple ways [50].

Several peptides have been identified as specific graphene binders by phage-display technology, opening a range of possibilities and challenges [51]. Indeed, the properties of graphene-based nanomaterials can be tuned by controlling structural and chemical parameters, such as thickness and, most importantly, degree of oxidation. The latter emerged as the main factor affecting surface chemistry and material organization. Therefore, graphene surfaces can be produced with different wettability and roughness by controlling the degree of oxidation. Protein adsorption can be predicted and modulated by the intertwining of these factors [52]. As an example, single-layer graphene oxide (GO) and few-layer graphene oxide can form smooth hydrophilic and anti-fouling surfaces, which significantly resisted unspecific protein adsorption. Conversely, low or non-oxidized graphene, due to its degree of roughness and hydrophobicity, promotes macromolecule binding. Furthermore, since peptides preferentially bind to the edge of planar graphene surfaces, the effects of graphene oxidation, number of layers, and underlying support were investigated using both experimental and computational methods [53].

These authors believe that carbon nanomaterials could inspire researchers toward intensifying in vitro studies on bare metal/metal oxide nanoparticles.

## 4. Molecular Biomimetics and Biomineralization

Biomineralization, the natural process leading to the biosynthesis of bones, dental structures, and mollusk shells, is a worthful example of the level of complexity and specificity that can be reached by the interaction between proteins and inorganic surfaces. This process shows the fine biological control exerted by living organisms over biomineral formation [54].

Thanks to their unique and specific interactions with small molecules, macromolecules and, even, inorganic materials, are smart building blocks controlling structures and functions of all biological hard and soft tissues in organisms. Regarding the interactions of biomolecules with abiotic materials, the recognition of surface features and defects is central to the absorption of peptides and proteins. On the surface of minerals such as calcite, the recognition and self-assembly of biomolecules on distinct surface kinks and steps can lead to changes in the overall shape and symmetry of the bulk crystal [55]. So and colleagues showed by in situ atomic force microscopy (AFM) that an acidic 20 kDa cement protein (MRCP20) from the barnacle *Megabalanus rosa* binds specifically to step edge atoms on {1014} calcite surfaces, and, with time, it further assembles into one-dimensional nanofibrils [55]. This selective surface interaction with step edge atoms directed a cooperative calcite modification, where templated protein nanostructures re-shaped the calcite crystal [55].

Combinatorial and directed evolution techniques led to the development of genetically engineered proteins for inorganics (GEPIs), which can specifically bind to selected inorganic materials. Combinatorial biological protocols, such as phage-display technologies, demonstrated the possibility of selecting peptide sequences with the ability to specifically recognize solid surfaces, similarly to natural proteins, regulating crystal growth in biomineralization [56]. As an example, Patwardhan and colleagues found sequence similarities among peptides strongly interacting with amorphous silica nanoparticles of various sizes (15–450 nm) [57]. Palafox and colleagues studied peptides selected as a function of their affinity for Ag and Au surfaces [58]. The authors combined experimental binding measurements, advanced molecular simulations, and selected nanomaterial synthesis, to show that interaction selectivity and recognition of metallic surfaces by peptides occurred by thermodynamically different binding modes [58]. In turn, the adsorption of three homo-tripeptides (His-His-His, Tyr-Tyr-Tyr, and Ser-Ser-Ser) on Au surfaces was investigated by Hughes and co-authors using molecular dynamic simulations [59]. The authors’ findings suggested that Au facets affected the structure and energetics of the adsorbed biomolecules, highlighting the possible impact in the field of the model used for the interpretation of experimental binding data [59]. Among noble metal nanoparticles, supposed to be characterized by similar surfaces, significant differences were reported in the binding selectivity of silver and gold nanoparticles toward peptides, depending on amino acid side chains. Tryptophan was singled out as the only aromatic amino acid with the ability to produce a tight binding on gold surfaces [60]. Differently, using the same type of peptides, AgNPs displayed no strong interaction with aromatic residues but showed high affinity for lysine side chains [61]. Such a degree of specificity between peptides and inorganic surfaces recalls the level of affinity existing among biological entities (i.e., antigen-antibody interactions).

Several human illnesses, such as Alzheimer’s disease, Creutzfeldt-Jacob disease, and dialysis-related amyloidosis, involve the formation of amyloid fibrils due to peptide aggregation. Fibril formation is a self-assembly process normally involving soluble peptides and proteins as building blocks that develop macroscopic structures with important connections with biological functions and human diseases. Fibril formation occurs by nucleation-dependent phenomena characterized by a lag-phase, leading to a critical seed produced from the free monomers in solution. This process represents the rate-determining step of the phenomenon after which the formation of amyloid fibrils proceeds rapidly [62]. Moreover, amyloid fibrils act as nucleation surfaces for new peptide and protein aggregation, leading to the autocatalytic growth of the fibril. These nucleation and elongation processes occur at distinct “sites” on the fibril: while nucleation seems most frequent along the sides of fibrils, elongation occurs at their ends. As nucleation leads to the formation of toxic oligomers, nucleation inhibitors were proposed for therapeutic purposes [63,64]. Importantly, the fibril formation process of amyloid proteins can be affected by the presence of foreign surfaces and profound effects in the modulation of amyloid formation by nanoparticles were reported [65,66,67]. In turn, the amino acid sequences have an effect on both fibrillization patterns and nanoparticle modulation envisaging the existence of specific interaction modes between particles and proteins [68]. Several types of nanoparticles, such as co-polymer particles, cerium oxide particles, carbon quantum dots, and carbon nanotubes, were found to enhance the rate of protein fibrillization. Linse and co-authors [65] explained the reduction of the lag phase in the presence of nanomaterials in terms of the increased probability of reaching a critical nucleation seed for protein fibrils. Moreover, the observed shorter lag (nucleation) phase in the presence of nanomaterials depends on the chemical nature of the particle surface. This nanoparticle-assisted nucleation mechanism may increase the risk of toxic clusters and amyloid formation. On the other hand, it paves the way to new routes for the controlled self-assembly of proteins and peptides and the production of novel nanomaterials [65]. A correlation between protein stability and aggregation propensity has been documented by Szczepankiewicz et al. [69] Moreover, Cabailero [68] and co-authors studied the development of amyloid fibrils by comparing a series of five mutants of the single-chain sweet protein monellin, differing for their intrinsic stabilities toward denaturation, and co-polymeric nanoparticles. A clear correlation between intrinsic protein stability and nanoparticles on the aggregation rate was reported and, in particular, mutants with a high intrinsic stability and low aggregation propensity showed an accelerated rate of amyloid fibril formation induced by the presence of nanoparticles. Conversely, nanoparticles led to a retardation of amyloid fibril formation in mutants characterized by low intrinsic stability and high aggregation rates. Moreover, both activating and inhibiting effects on amyloid fibril formation were particularly pronounced in the presence of hydrophilic nanoparticles, which presented a large surface accessibility for hydrogen bonding on the polymer backbone [68].

The selectivity of bare iron oxide nanoparticles for protein binding was exploited for an application in biomedicine. Naked nanostructured iron oxide with selectivity toward proteins was tested in vivo on zebrafish [70]. The nanomaterial was able to specifically bind Apolipoprotein A1 (Figure 2, left) and to evade the clearance of the immune system of zebrafish by mimicking, to some extent, high-density lipoproteins (HDL). The structural analogy with HDL was also corroborated by the massive absorption of the nanomaterial by the intestinal tract and, most importantly, by the presence of the nanomaterial in the host ovaries (Figure 2, right). The tissue-specific delivery of an antibiotic and the absence of an adverse outcome were demonstrated, substantiating the idea that the stealth effect is a consequence of the formation of a shell of selectively bound functional proteins on the nanomaterial, and opening a new avenue to the evaluation of metal oxide nanoparticles as novel stealth nanomaterials for biomedicine.

## 5. Prebiotic Chemistry and Chirality Selection

Metal oxide surfaces are believed to have fulfilled a central role in the assembly of early macromolecules and, therefore, of life. Nevertheless, these materials are not sufficiently studied as models on the specificity of the interactions between proteins and inorganic materials.

Metal oxide-based bioinorganic hybrids offer unique structures, physicochemical features, as well as novel biochemical properties. In the last decade, extensive research efforts were dedicated to TiO_2_, ZnO, SiO_2_, and GeO_2_ metal oxides, aimed at improving methods of synthesis and surface functionalization of these nanomaterials, as well as at their shaping and structural patterning, taking inspiration from natural processes. As an example, the binding of the protein tyrosinase on birnessite, a manganese oxide mineral, showed very high affinity. The enzyme molecule was not intercalated or adsorbed in the mineral, but was immobilized on the external surface of the metal oxide [71,72].

Again, metal oxide surfaces are believed to have played an essential role in triggering the generation of biomolecules and, therefore, life, possibly by participating in the concentration of biomolecules from dilute solutions, as well as at their organization into structured biopolymers [73,74,75,76,77,78,79]. Cairns-Smith’s group proposed the involvement of defective clay crystals as the first “genetic” code (i.e., not based on nucleic acids) fulfilling an essential function during the early stages of life evolution [80]. Moreover, it was shown that metal complexes were sequestered from metal oxide surfaces by small organic molecules (e.g., polypeptides) to form the precursors of active centers of enzymes. Russell’s group proposed that iron oxides could bind proteins to form catalytic entities, possible precursors of oxygenic photosynthesis [81]. On the other hand, it was proposed that mineral surfaces can recruit and select peptides from their environment leading to novel macromolecules with improved catalytic activities [82]. Thus, minerals had an important role in prebiotic chemistry, and they can be thought of as templates, promoting structural organization in disordered organic molecules. An ordered structure can be induced in a peptide containing a suitable amino acid sequence by a surface providing properly spaced reactive sites. Lundqvist and colleagues demonstrated this hypothesis by showing that a disordered peptide can be forced into a well-defined structure by silica nanoparticles [83]. In another example, a peptide displaying a low helical content in solution was induced to adopt a well-defined α-helix upon interaction with silica nanoparticles [84]. Another possibility inspired by biological processes was the development of silica-based nanoparticles as smart matrices for the auto-encapsulation and controlled release of functional molecules (e.g., proteins). This approach involved the use of silica-forming peptides to mediate the in vitro generation of silica hybrids as potential therapeutic nanocarriers [85].

For many decades, the origin of homochirality in biological systems has aroused the interest of the scientific community, and it has been in a core position in studies on the origin of life [86]. The central question deals with how enantiomeric separation could take place in the absence of a chiral symmetry operator. In this view, one fascinating feature of metal oxide surfaces is that they can provide periodic environments prone to select, concentrate, and possibly even organize chiral molecules into polymers and other macromolecular structures [87]. In this field, several phenomena were identified, including solid surface reconstruction, chiral imprinting upon adsorption of biomolecules, and the enhancement or suppression of enantio-selectivity of surfaces in the presence of racemate mixtures of chiral compounds [88]. Naturally occurring clay surfaces were described to selectively adsorb amino acids and act as chiral amplifiers, as reported for vermiculite clay, prompting the involvement of non-centrosymmetric chiral surfaces at the origin of biological homochirality [89]. Additionally, the role of clays was already proposed for RNA oligomerization [90] and in the organization of amphiphilic molecules and lipids on mineral surfaces [91], further suggesting the importance of these minerals in the origin of chirality in lagoonal environments. In this view, naturally occurring abiotic surfaces can inspire the synthesis of novel nanostructures, which are usually highly symmetric. Not surprisingly, in the last decades, a number of chiral nanostructures were reported in the literature [92]. As an example, hints on the dichroic behavior of iron oxide nanoparticles were provided [93]. Chiroptical activity, commonly considered a prerogative of tetrahedral organic molecules, was an unprecedented feature for iron oxide nanoparticles. Crystalline vacancies on the nanomaterial surface were identified as chiral centers and studied in the presence of inorganic ligands. The dichroic signal of nanoparticles was differently influenced by the coordination of different chelators (Figure 3, left) and this phenomenon was attributed to the chelator ability to rescue defective crystalline sites on the nanomaterial surface. Furthermore, ligand distribution was strictly governed by the lattice of iron oxide nanoparticles (Figure 3, right).

## 6. Protein and Peptide Purification by Bare Nanomaterials

As highlighted above, material selectivity toward protein binding represents an advantageous feature that can be exploited for the development of effective biomedical applications of nanomaterials. Some nanoparticles came into the limelight for being extremely selective and were applied for the isolation and purification of proteins in complex biological matrices.

Immobilized Metal Ion Affinity Chromatography (IMAC) uses metal ions such as Ti^4+^, Fe^3+^, Ga^3+^, Al^3+^, and Co^2+^ immobilized by a linker to a solid matrix and it has become one of the most widely applied techniques for the selective enrichment of phospho-peptides and phospho-proteins [94,95,96,97]. IMAC has been successfully applied to large-scale phospho-proteome studies, but it still lacks satisfactory selectivity for separating mono-, di-, and multi-phosphorylated peptides [98]. In this context, metal oxide materials, such as titanium dioxide (TiO_2_), zirconium dioxide (ZrO_2_), iron oxides (hematite and maghemite, Fe_2_O_3_, or magnetite, Fe_3_O_4_), and alumina (Al_2_O_3_), have recently been proposed as alternatives to IMAC to specifically separate phospho-peptides from complex samples [99]. This application was defined as Metal Oxide Affinity Chromatography (MOAC) [100]. Among these purification strategies, TiO_2_ is the most commonly used metal oxide for the selective capture of target peptides because it is extremely tolerant toward most buffers and salts used in biochemistry and cell biology [101,102,103,104]. TiO_2_ has been proven to exhibit high affinity and good selectivity for binding phospho-peptides for mass spectrometry [104,105,106]. Although the exact physicochemical mechanism is largely unknown, several additives, such as aromatic and aliphatic hydroxy-carboxylic compounds, have been reported to reduce nonspecific binding on TiO_2_ leading to dramatic enrichment improvements [107]. Alternatively, tin dioxide (SnO_2_) nanoparticles were successfully synthesized and applied to selectively purify phospho-peptides for mass spectrometry analysis [108]. Ma and colleagues showed that octahedral SnO_2_ nanoparticles, characterized by abundant under-coordinated Sn atoms, exhibited high affinity and selective binding ability for phospho-peptides [108]. Thus, low-abundance phospho-peptides were selectively and efficiently purified from complex biological samples. Most importantly, the authors tailored the exposed facets of the nanomaterial for modifying its physical and chemical properties and for improving its binding selectivity [108].

Superparamagnetic Iron Oxide Nanoparticles (SPIONs) constituted of magnetite (Fe_3_O_4_) or maghemite (γ-Fe_2_O_3_) have proved their usefulness in many biomedical applications, such as drug delivery, induced-hyperthermia, and as MRI (magnetic resonance imaging) contrast agents [109]. Their wide use is mainly due to the favorable biocompatibility and their magnetic properties. Moreover, the use of bare iron oxides offers advantages for industrial applications, mainly due to the low-cost and rapid synthesis. Generally, depending on the specific application, several surface coatings have been used for stabilizing colloidal suspensions and for tuning the properties of these nanoparticles [110]. Several alternatives have been proposed for the surface modification of magnetic nanoparticles, such as coating with metallic gold, organic polymers, and silica [110,111,112]. Nevertheless, the introduction of a coating layer presents several drawbacks. Indeed, stabilizing coatings can bind weakly to nanoparticle surfaces and eventually desorb or exchange with bulk solution, affecting the stability of colloidal suspensions. In addition, coating processes are often time-consuming, expensive, and characterized by low yield, limiting massive productions. Furthermore, the coating reduces the magnetic moment of the nanomaterial by introducing a non-magnetic component. At last, these coatings strongly modify the nanomaterial surface and influence the interactions with proteins in the biological environment.

Blank-Shim and colleagues performed the first systematic study on the interactions between peptides and bare iron oxide nanoparticles, aimed at developing peptide tags to be genetically engineered into proteins for the recognition of nanomaterials [113]. The strategy for designing high-affinity peptide tags required an in-depth understanding of the surface-peptide recognition patterns [56]. A model considering a patchwork of electrostatic sites on the iron oxide surface was used to describe the adsorption affinity of all 20 natural amino acids [56]. Results indicated that the binding efficiency of peptides can be fully explained by electrostatic interactions, providing the basis for the design of peptide tags for biomolecule recognition of bare magnetic nanoparticles, which was experimentally demonstrated with tagged Green Fluorescent Protein (GFP) variants [56]. The study paved the way for a fast and simple protein immobilization procedure without the need for unstable and expensive affinity ligands, currently in use for the isolation of recombinant proteins [113].

A tool for the prediction of the propensity of peptides to interact with iron oxide nanoparticles was presented by Schwaminger and colleagues [114]. The authors reported on an effective implicit surface model (EISM) considering electrostatic interactions, van der Waals interactions, and entropic effects for the theoretical calculations. However, the most important parameter was a force field contribution term of the surface accessible area directly derived from experimental results on the interactions of nanomaterials and peptides. EISM was verified by further peptide binding experiments in an iterative optimization process where negatively charged peptides were identified as the best binders for iron oxide nanoparticles. Besides electrostatics, bare iron oxide nanoparticles revealed the ability to effectively isolate His-tagged proteins from complex biological matrices [115]. This property can be attributed to the availability of under-coordinated iron(III) sites on iron oxide surfaces as a consequence of dangling bonds derived from crystal truncation [116]. In this view, the reactivity and binding selectivity of the uncoated iron oxide surface can be explained by the behavior of iron(III) sites and their topographical distribution on the nanoparticle surface. A comparison using different model proteins revealed the existence of patterned regions of carboxylic groups acting as recognition sites for naked iron oxide nanoparticles. Readily interacting proteins display a distinctive surface distribution of carboxylic groups, recalling the geometric shape of an ellipse (Figure 4B,F). This was interpreted as a morphological complementarity of nanoparticle curvature, compatible with the topography of exposed iron(III) sites laying on the iron oxide nanomaterial surface [117]. The macromolecule recognition site, absent in non-interacting proteins, promoted the harboring of the protein on the nanomaterial surface and allowed the formation of functional protein coronas.

Aiming at the diagnosis of threatening pathologies, such as cancer, the analysis of protein recognized by nanomaterials represents an advantageous diagnostic strategy, as it can simplify the analysis of the whole sample proteome [118,119]. Proteome variations following the occurrence of a disease can be reflected by an alteration of the protein corona composition in comparison with healthy controls. In this view, bare iron oxide nanoparticles were successfully employed for the discrimination between milk coming from healthy and mastitis-affected bovines [120].

## 7. Surface Energy and Protein Corona

Surface free energy is commonly considered the key factor for explaining the size-dependent thermodynamic behavior of nanoparticles. Indeed, at the nanometer dimension, nanomaterial structures and properties are governed by surface effects. This is due to the nanomaterial’s high surface-to-volume ratio, resulting in the contribution of a significant fraction of the nanostructure atoms to surface phenomena with respect to the bulk. As the size of nanomaterials decreases, the ratio of surface atoms involved sharply increases.

The environment of surface atoms is different from the atoms in the bulk. In the interior, each atom is surrounded in all directions by neighbor atoms, whereas surface atoms present an anisotropic environment. Thus, no net force is exerted on the bulk atoms, while surface atoms are subjected to an asymmetric force field, resulting in a higher energy level associated with the surface. This excess energy at the surface is named surface energy [121]. Surface energy is fundamental for understanding nanoparticle thermodynamics, hence, different approaches were proposed that were aimed at the prediction of this quantity, including classical thermodynamics calculations, molecular dynamics simulations, and ab initio calculations [122]. Even if the concept of surface energy is a hot topic in several applications, such as energy storage and catalysis, it can, as well, be extremely useful for the comprehension of the interactions between, for instance, proteins and bare inorganic nanoparticles. The lowering of surface energy is a natural process and, from a thermodynamic standpoint, protein corona formation can be considered as a result of the tendency of nanoparticles to minimize their surface energy in the presence of biological macromolecules [123,124]. Indeed, the net change of Gibbs free energy following protein binding on the nanoparticle surface is negative, showing that the process occurs spontaneously [125]. Protein corona formation competes with nanoparticle aggregation, which lowers surface energy, as well, by reducing the surface-to-volume ratio [126].

Regarding bare crystalline nanomaterials, the reduction in surface free energy is associated with surface reconstruction, namely to the re-organization of the lattice at the boundary with the solvent, which, in turn, can induce conformational and functional changes of bound proteins [28]. In this view, the surface energy of nanocrystals depends on the high density of under-coordinated atoms at surface steps and kinks. It is well-known that these features can determine exceptional catalytic properties [127]. In recent decades, intensive efforts were dedicated to the synthesis of tailored crystalline materials with different properties ascribed to differently exposed atoms at the material surfaces [128,129,130]. It should be considered that a plethora of structurally unprecedented motifs have been recently developed, including prisms, polyhedrons, rods, plates, wires, and so on, and the list of available shape-guiding synthetic processes is constantly increasing [131]. In this context, surface energy excess and distribution are important topics to understand the growth, reactivity, and stability of materials on the nanometric scale [127,132]. Chemical species preferentially interact with specific crystal planes as a result of the different surface energies [133].

As an example, a systematic investigation of the surface energy of cubic nanoparticles was carried out using the modified embedded atom method (MEAM) [134]. Results showed that both surface energy and dangling bond density increased with decreasing material size [135]. Accordingly, Holec and colleagues observed that surface energy was correlated to the number of dangling bonds (reduced coordination of the surface atoms) [136]. Ferrer and colleagues proposed a morphology diagram reporting the quantitative energy distribution of nanomaterials, that can be used as a guide to understand the relationship between crystal growth and surface interactions and to help experimental work [137]. Ma and Xu (2007) proposed a quantitative calculation of the surface energy of nanomaterials [135]. Furthermore, a recent study by Visalakshan and colleagues stressed the importance of taking nanoparticle shape into serious consideration as it can significantly influence protein corona formation and the therapeutic potential of nanoparticles [138]. Indeed, authors reported that different surface energies are correlated with different nanoparticle shapes, in agreement with the previous literature [139,140]. A reachable goal should be the translation of shape-guiding synthetic processes into the development of novel bare nanomaterials for biomedical applications.

Energy distribution on a nanoparticle surface can be meant as a key aspect for the interpretation and prediction of the selectivity toward protein binding. In the absence of an optimal complementarity at the interface between the nanoparticle and the macromolecule, the binding is thermodynamically unfavored. On the contrary, if the two surfaces suitably match, the surface energy of the nanomaterial is strongly reduced, resulting in nanoparticle stabilization, and the binding with the specific macromolecule is favored. Thus, ideally, nanoparticles are designed for interacting with specific proteins. It is important to bear in mind the strong link existing between shape-controlled synthesis, surface energy, and surface coordination chemistry. Nevertheless, the bottleneck in the research on the surface coordination chemistry of nanomaterials is mainly related to the lack of effective tools to characterize coordination structures at surfaces [141]. With the comprehension of surface coordination chemistry, the molecular mechanisms behind various important effects of inorganic nanomaterials can be disclosed, including their actual potential for biomedical applications.

## 8. Conclusions

The creation of protein-like nanoparticles represents the Holy Grail in nanoscience, where the crucial challenge is to endow nanoparticles with protein-like specificity and enabling the abiotic material to interplay with biological systems [142,143]. In this view, as nanoparticles and proteins are entities of the same size, protein–protein interactions could inspire our comprehension of the selectivity between biological interfaces and synthetic, abiotic objects. Although the recognition of peptides and proteins by bare inorganic nanomaterials is at the core of a range of scientific branches, its potential in biomedicine is far from being fully exploited. Possibly, this could be due to the limitations of many reported synthetic inorganic nanomaterials, which do not present suitable characteristics to be used in the absence of coating materials or surface stabilizers. On the contrary, novel syntheses could bring to light unexpected and fascinating surface chemistries that would merit the attention of the biomedical community.

The central question remains: how an abiotic surface of a generic inorganic nanomaterial can be endowed with a protein-like specificity? We believe that the answer could be hidden in the enormous body of studies related to, besides nanotechnology and nanomedicine, prebiotic chemistry, biomineralization, and chiral nanomaterial selectivity dealing with specific and sophisticated biotic-abiotic interactions (Figure 5). As an example, it is important to mention that the success of biomedical implants depends on the interaction between implant surfaces and the biological environment and, in this view, it is well accepted that nanotopography is a key factor determining protein adsorption, as well as cell growth onto a given biomaterial and, therefore, its biocompatibility [144].

Not surprisingly, phage-display technology, a technique largely employed for the study of protein–protein, protein–peptide, and protein–DNA interactions, identified several peptides as specific binders of naked abiotic material surfaces [45]. Furthermore, computational methods have become increasingly reliable to understand the recognition mechanisms at inorganic–biological interfaces [145]. Based on rational design and molecular dynamics simulations, a relatively simple computational method for screening protein–nanoparticle interactions was provided by Penna and Yarovsky [146]. Noteworthy, this method considered the entropic penalty due to protein binding as a loss of ligand flexibility and interfacial water trapping, resulting in a restriction of macromolecule adsorption. In this view, specific surface topographies and surface modifiers would determine protein coronas with selected compositions [147]. Regarding protein orientation and structural evolution upon corona formation, Bourassin and colleagues proposed a coarse-grained, elastic network representation to implicitly model the impact of surface adsorption on protein mechanics [148]. However, to obtain design rules for proteins and surfaces with determined binding characteristics, researchers often oversimplify the interaction mechanisms, focusing on only one or a few of the properties influencing the affinity or the specificity of the proteins for the surfaces. The transferability of simple design rules developed in one study to other systems, under different conditions, has to be taken with caution [149].

Intriguingly different nanoparticles can display an affinity for specific amino acid side chains, as in the case of CNTs and AuNPs, both displaying selectivity for peptides characterized by tryptophan-rich sequences [46,60]. However, the amino acid sequence of peptides involving specific secondary structures and the surface distribution of functionalities of proteins exert control over the interaction with nanoparticles.

As a general concept, nanoparticle–protein interactions would follow a scheme involving a collection of different complementarities and affinities between the two surfaces, each one displaying a specific topography of functional groups and binding moieties, as well as shape complementarity [47]. One opportunity for improvement is to overcome the difficulty to describe and represent the surface of naked inorganic nanomaterial with the required details. The surface atomic layer of a solid is a discontinuity point, an interface between the bulk structure (either crystalline or amorphous) and the surrounding medium (air, water, or solvents). The surface of a solid is a complex and dynamic entity, that can actively interact with the surrounding medium, e.g., undergoing protonation/deprotonation, adsorbing/desorbing solvent molecules, ions, and biomolecules, or assisting the precipitation of compounds [150]. In this view, and taking inspiration from chemo-informatics, the future possibility to encode multi-component structures of nanomaterials in a machine-readable format was foreseen [151]. Information on size, shape, internal structure, and surface characteristics, possibly including ligands and surface defects on uncoated nanomaterials, should be considered.

Finally, the present review suggests the presence of a common trait among different bare surfaces displaying specificity for proteins. This is represented by specific patterns, namely distributions of chemical groups, defects, and facets on nanomaterials. It is the authors’ opinion that a crucial task for the successful application of these nanomaterials will be the development of approaches for mapping surfaces and, as a consequence, identifying specific distributions of chemical functionalities on surfaces that are able to mimic biomolecules (Figure 5). Furthermore, taking inspiration from studies on carbon nanomaterials, bare inorganic nanomaterials displaying selectivity toward protein corona formation should be an object of more intense in vitro evaluations. In contrast with the body of literature highlighting the level of selectivity that can be reached by this class of nanomaterials, their in vitro behavior lacks an adequate examination. This will be of crucial importance for deepening the comprehension of the relationship between surface recognition/protein envelope/cell responses, as well as for the enrichment of the biomedical field with novel potential theranostic tools. Concluding, this approach would start a new era in nanomedicine, bridging the gap between material science, nanomaterial synthesis, structural biochemistry, biophysics, and biomedical applications in real-world scenarios.

## Figures and Tables

**Figure 1 ijms-22-07625-f001:**
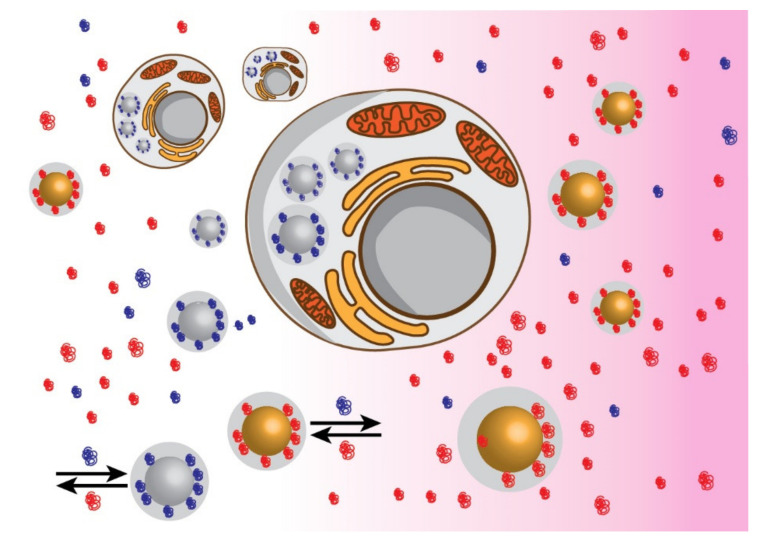
In the schematic representation, upon exposure to a biological milieu, most of the nanoparticles reported in the literature (yellow) are subjected to the unspecific binding of proteins (red). On the contrary, nanoparticles displaying binding selectivity (grey) lead to the formation of a shell composed of specific proteins (blue) promoting cell internalization. This biological envelope makes the difference between nanoparticles being internalized or being cleared by the immune system. Arrows represent the Vroman effect.

**Figure 2 ijms-22-07625-f002:**
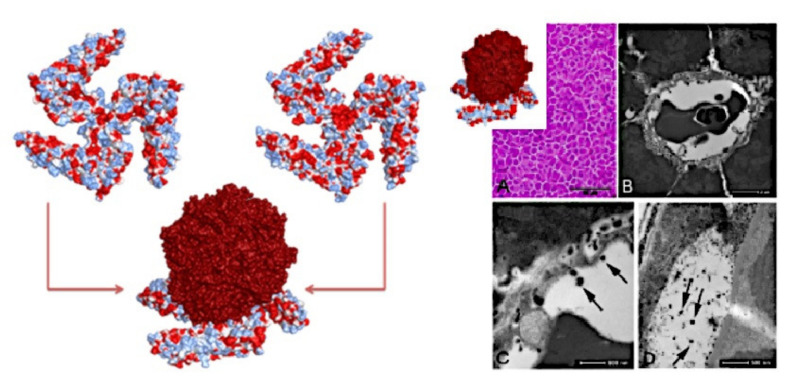
Apolipoprotein A1 recognition of nanoparticle surface (left) and electron microscopy images of biological samples showing nanoparticle uptake in Zebrafish (right): (**A**) photomicrograph of a liver parenchyma cross-section showing the organization of hepatocyte cells (hematoxylin and eosin staining); (**B**) transmission electron microscopy image of hepatocyte cells surrounding a capillary; (**C**) high magnification of the region shown in (**B**) revealing nanoparticles (arrows) inside the capillary; (**D**) transmission electron microscopy image of nanoparticles in the space between the follicular epithelium and zona radiata of the zebrafish ovary (reproduced with permission from [70]).

**Figure 3 ijms-22-07625-f003:**
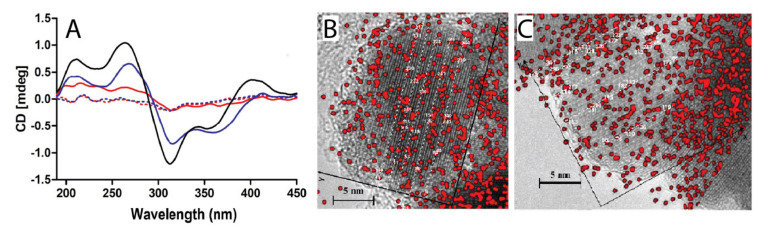
Circular dichroism (CD) and EDS chemical mapping on HR-TEM micrographs of arsenic (As(III) and As(V)) coated iron oxide nanoparticles. Panel (**A**): CD spectra of unmodified nanoparticles (black continuous line), As(III) or As(V) modified nanoparticles (red and blue continuous lines, respectively), and As(III), As(V) as controls (red and blue dashed lines, respectively). Panel (**B**) and (**C**): mapping of nanoparticle surface sites displaying chiroptical activity: the distribution of As(III) (panel (**A**)) and As(V) (panel (**C**)) oxyacids from EDS chemical mapping on HR-TEM micrographs. Segmented white lines represent the (220) planes of iron oxide crystalline lattice (reproduced with permission from [93]).

**Figure 4 ijms-22-07625-f004:**
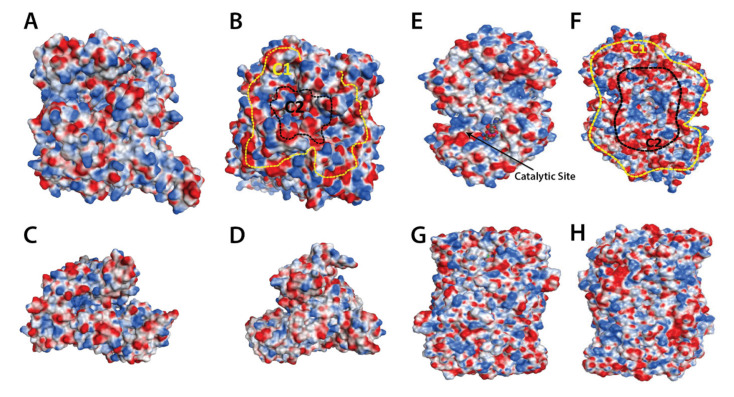
Three dimensional computational models of proteins displaying selectivity toward iron oxide nanoparticles (A,B,E,F) and proteins with no affinity for iron oxide nanoparticles (C,D,G,H). Patterns of carboxylic groups can be visualized on specific sides of the macromolecules. (**A**,**B**) = xanthine oxidase (XO); (**C**,**D**) = bovine serum albumin (BSA); (**E**,**F**) = aminoaldehyde dehydrogenase 1 from tomato (SlAMADH1, from Solanum lycopersicum); (**G**,**H**) = bovine serum amine oxidase (BSAO) (reproduced with permission from [117]).

**Figure 5 ijms-22-07625-f005:**
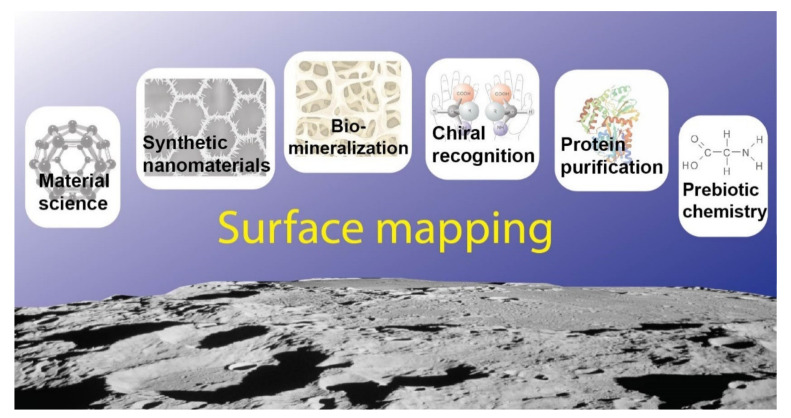
Surface mapping emerges as the key to understanding the high level of complexity and specificity that can be obtained in the interaction between proteins and bare inorganic surfaces.

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
