# Peer review of "Toward the Specificity of Bare Nanomaterial Surfaces for Protein Corona Formation"

_ijms, 2021, doi:10.3390/ijms22147625_

Round 1
Reviewer 1 Report
Original comments I read this paper word by word and my decision is based on the aims of this reiew papers and conclusion with suggestions. I believe the authors tried to cover most of the things within the limit as per journal policy. However, I do think that this review lacks many things related to protein corona formation such as a general scheme to present the protein-corona complex formation and interaction with cells. If we want to see this review paper more critically, then this topic is already reviewed by many and there is nothing new this review paper (except update information >15%). But the content of this review paper is truly described and an interesting review indeed, which dealt with, pristine nanomaterial surfaces for
protein corona formation . I would recommend accepting this paper in the current version.
------
Revised comments: Accepted in current format.
Reviewer 2 Report
The submitting author states that this manuscript is not a review of the protein corona, despite the fact that the two reviewers and the editor took it to be exactly that. This new version is simply the older one, with several additions. As both reviewers previously pointed out, it is limited, and contains nothing new.
I consider it to be a review that covers a limited area of protein corona formation, which will attract a limited audience. Such a review has limited value in being published. However, publication would probably aid those interested, by helping them carry out their own abstracting. Thus, I cannot stand in the way of its publication, although I step aside warily.
I suggest that, first, the text be reread by a native English speaker who is familiar with the subject. Some of the phraseology does not make sense to me. I give two examples, both in the Abstract:
- Lines 15 and 16: “..was revised.” doesn’t make sense. Do the authors mean “.. is reviewed.”?
- Last sentence: I do not understand what the authors are saying, or whether it is necessary to say it.
Other confusing phraseology exists throughout the text.
